# *Clostridium difficile* Colonization in Oncologic Patients Undergoing Major Abdominopelvic Surgery: To Treat or Not to Treat? An Observational Study with Propensity Score Analysis

**DOI:** 10.3390/medicina61091606

**Published:** 2025-09-05

**Authors:** Sorinel Lunca, Wee Liam Ong, Raluca Zaharia, Romulus Mihaita Pruna, Gabriel Mihail Dimofte, Stefan Morarasu

**Affiliations:** 1Department of Surgery, Grigore T Popa University of Medicine and Pharmacy, 700115 Iasi, Romania; sdlunca@yahoo.com (S.L.); raluca.zaharia11@yahoo.com (R.Z.); gdimofte@gmail.com (G.M.D.); morarasu.stefan@gmail.com (S.M.); 22nd Department of Surgical Oncology, Regional Institute of Oncology (IRO), 700483 Iasi, Romania; 3Department of Infectious Diseases, Regional Institute of Oncology (IRO), 700483 Iasi, Romania; romulus_pruna@yahoo.com

**Keywords:** *Clostridium difficile*, oncology, abdominopelvic surgery, cancer, antibiotics, morbidity

## Abstract

*Background: Clostridium difficile* colonization (CDC) represents a clinical concern in oncology patients undergoing abdominopelvic surgery, particularly regarding the potential role of prophylactic antibiotics in preventing progression to active infection. *Methods:* We performed a single-center, retrospective, case-matched observational study of oncology patients with CDC who underwent abdominopelvic surgery between 2018 and 2023. Patients were divided into two cohorts: those who received prophylactic antibiotics and those who did not. Postoperative outcomes were compared using propensity score matching (PSM). Logistic regression and ROC curve analysis were applied to assess the predictive value of antibiotics relative to other comorbidities. *Results:* Ninety patients were included (62 with antibiotics; 28 without). In the unmatched cohort, patients receiving antibiotics showed a non-significant trend toward higher morbidity (32.2% vs. 21.4%, *p* = 0.327) and surgical site infection rates (9.6% vs. 0%, *p* = 0.171). After PSM (26 patients per group), morbidity remained comparable between cohorts (30.7% vs. 23.0%, *p* = 0.538). Notably, no patient developed active *C. difficile* infection during follow-up, regardless of antibiotic use. Antibiotic therapy was not an independent predictor of postoperative morbidity (OR 1.746, *p* = 0.297; AUC 0.549, 95% CI 0.405–0.687). *Conclusions*: In this study, prophylactic antibiotic use in CDC patients undergoing abdominopelvic oncology surgery was not associated with improved postoperative outcomes. While no progression to active infection was observed, the potential benefits of prophylaxis remain uncertain. Larger, prospective studies are needed to clarify the clinical role of antibiotics in this setting.

## 1. Introduction

*Clostridium difficile* infection (CDI) is a leading cause of healthcare-associated infectious diarrhea and is associated with significant patient morbidity [1]. Differentiating between *Clostridium difficile* colonization (CDC) and active infection (CDI) is clinically important. Colonization is defined as a positive *Clostridium difficile* test—with or without toxin detection—in the absence of clinical symptoms such as watery diarrhea, fever, anorexia, nausea, or abdominal pain. In contrast, CDI refers to symptomatic patients with a positive result [2].

Numerous studies have demonstrated that cancer patients with CDI experience significantly higher rates of morbidity and mortality [3]. Oncology patients are particularly susceptible to CDI due to several contributing factors, including prolonged hospital stays, frequent exposure to broad-spectrum antibiotics, immunosuppressive chemotherapy regimens, and invasive surgical procedures [4,5,6]. Furthermore, research has shown that patients with hospital-acquired acute CDI (HA-CDI) are at significantly greater risk of developing postoperative complications—such as anastomotic leakage—than *Clostridium difficile* colonization (CDC) patients [7].

A specific clinical dilemma arises when oncological patients, particularly those undergoing abdominopelvic surgeries, are found to be colonized with *Clostridium difficile* (*C. Difficile*). Should these patients receive prophylactic antibiotics to prevent progression to active infection? Although many studies have reported increased morbidity and prolonged hospital stays in patients with acute *C. difficile* infection (CDI) [8,9,10], there is a lack of research focusing on oncology patients undergoing abdominopelvic surgery (e.g., gastrointestinal surgery, gynecology surgery, and hepato-pancreato-biliary surgery) who are colonized but not infected with *C. difficile*. Traditionally, standard practice has been not to treat asymptomatic carriers, under the rationale that treating a carrier confers little benefit (since the patient is well) and may even be harmful (by eradicating the carrier strain only to predispose the patient to acquisition of a new, potentially more virulent strain, or by inducing antimicrobial resistance) [11]. This conservative approach is reflected in most guidelines, which recommend against testing or treating patients for *C. difficile* in the absence of CDI symptoms, citing lack of durable benefit, risk of microbiome disruption, antimicrobial resistance, and potential loss of natural colonization resistance [12,13]. However, emerging data and clinical scenarios have prompted re-examination of this stance. In high-risk populations, for instance, patients with cancer who are about to receive intensive chemotherapy or undergo an operation involving bowel anastomosis or major oncological abdominal surgery, some clinicians wonder if preemptive treatment of a known carrier might avert a postoperative CDI or its complications (such as anastomotic leak). The stakes are indeed high: postoperative CDI has been associated with doubled rates of anastomotic leakage and surgical site infection in colorectal cancer surgery [7], and in gynecologic oncology patients, CDI can disrupt chemotherapy cycles and even lead to sepsis [14]. These serious outcomes raise the allure of any possible prevention strategy. In the STOP-CDI trial [15], patients in an oncology/transplant setting were screened for *C. difficile* on admission, and those colonized were given 10 days of oral vancomycin prophylactically. The outcome was striking: the incidence of hospital-onset CDI dropped dramatically (from ~5.6% to <1% overall, and in oncology patients from 7.5% to 1%). They also did not observe an increase in vancomycin-resistant enterococcus or other adverse events in the prophylaxis group.

To date, no study has specifically evaluated morbidity and mortality in this subgroup. This study aims to address the gap by examining postoperative complications and morbidity in oncology patients undergoing abdominopelvic surgery who are colonized with *C. difficile,* whether or not they receive antibiotic treatments.

## 2. Materials and Methods

### 2.1. Design and Setting

This is a single-center, single-department, observational, retrospective comparative study conducted on oncological surgical patients diagnosed with *C. difficile* colonization (CDC) at our institution between 2018 and 2023. All patients underwent standard oncological evaluation and treatment following multidisciplinary tumor board recommendations and were managed exclusively at our institution.

All patients were screened for *C. difficile* upon admission using a rapid stool test (CerTest Biotec^®^, Zaragoza, Spain). This method, however, has inherent limitations related to both detection performance and sensitivity thresholds. The manufacturer reports a sensitivity of 96.6% and a specificity of 99.4% for the assay [7]. Nonetheless, as acknowledged by the producer, test performance may be reduced in asymptomatic carriers or in cases of low-level colonization. Importantly, the reported validation data are based on symptomatic *C. difficile* infection (CDI), and no performance data are available for asymptomatic carriers.

For the purposes of our study, *C. difficile* colonization was defined as a positive admission test result, irrespective of toxin detection. Patients were not routinely re-tested unless they developed symptoms, such as ≥3 stools per day or watery diarrhea. Those who subsequently tested positive following the onset of symptoms were classified as having acute *C. difficile* infection (CDI).

Patients that tested positive for *C. difficile* (GDH [Glutamate Dehydrogenase]-positive and toxins A and B positive) were treated according to our institutional protocol aligned with international guidelines. This included patient isolation, contact precautions, and oral Vancomycin (125 mg QDS), with dose escalation as needed. Patients that tested positive for *C. difficile* (GDH-positive and toxins A and B negative), defined as CDCs (*Clostridium difficile* carriers), were selected for further analysis depending on whether they received antibiotic treatment or not. The decision to initiate perioperative antibiotic therapy (oral Vancomycin, 125 mg QDS) was made by the attending consultants’ preference considering the higher risk of postoperative *C. difficile* exacerbation. The adherence to the antibiotic prophylaxis protocol was verified by retrospectively analyzing the drug Kardex of each patient. No patients received systemic preoperative antibiotics beyond standard prophylaxis.

This study received ethical approval from our institutional ethics committee (registration number: 271/26 May 2025). Informed consent was waived as the research involved anonymized data and posed no risk to patients.

### 2.2. Inclusion and Exclusion Criteria

This study adhered to the STROBE checklist (Figure 1). A database of all patients tested for *C. difficile* at our institution was reviewed. Patients from other departments were excluded. Only those admitted to our department who underwent abdominopelvic surgical intervention were included. Exclusion criteria included patients with benign disease, those undergoing non-abdominal procedures (e.g., mastectomy, lower limb tumor ablation), and those undergoing minor interventions (e.g., esophageal stent placement, port-a-cath insertion).

### 2.3. Data Analysis

Clinical and surgical data were retrospectively extracted from electronic medical records. *Clostridium difficile* colonization (CDC) patients were stratified into two cohorts: the study group, CDC patients who received antibiotic treatment, and the control group, CDC patients who did not receive antibiotic treatment.

Collected variables included demographic characteristics, comorbidities, primary diagnosis, type of surgical procedure, length of hospital stay, antibiotic exposure, intraoperative complications, pre- and postoperative laboratory values, overall medical and surgical morbidity, and both in-hospital and 30-day mortality.

Quantitative variables were analyzed using independent sample *T*-tests, while categorical variables were analyzed using Chi-square tests. Odds ratios (ORs) with corresponding 95% confidence intervals (CIs) were calculated to assess the association between antibiotic use and postoperative complications. A *p*-value of less than 0.05 was considered statistically significant. To minimize potential confounding and selection bias, as was previously done, propensity score matching (PSM) was performed [16,17]. Matching control patients in the study group were selected according to propensity scores (PSM), in a 1:1 ratio, with patients in the control group based on age and comorbidities (diabetes, cardiovascular disease, renal disease). Covariate balance between treatment groups was evaluated using standardized mean differences (SMDs) computed with the pooled standard deviation, and displayed in a Love plot (cobalt package, R). Adequate balance was defined a priori as |SMD| < 0.10, with values shown before and after matching. Propensity score matching was performed 1:1 using Mahalanobis distance with a caliper of 0.20 and without replacement (Figure 2). ROC curves were made to assess to what extent morbidity can be predicted when comparing CDC that received/did not receive antibiotics versus age and other comorbidities. As a sensitivity analysis, we fitted a multivariable logistic regression in the unmatched cohort with overall postoperative morbidity (0/1) as the outcome. The model included prophylactic antibiotic use (yes/no), age (years), diabetes mellitus, cardiovascular disease, and chronic kidney disease. Adjusted effects are reported as adjusted odds ratios (aORs) with 95% confidence intervals (CIs). Discrimination was summarized by the area under the ROC curve (AUC) (95% CI via bootstrap), and calibration by the Hosmer–Lemeshow test (g = 10). Variance inflation factors (VIFs) were inspected to screen for multicollinearity. Statistical analyses were performed using JASP (version 0.19.3). PSM was performed via MedCalc v23.3.5. Balance diagnostics and visualization (SMD table and Love plot) and the sensitivity logistic regression were implemented in R (cobalt/dplyr/ggplot2).

Key: Absolute standardized mean differences (SMDs) for each covariate are shown pre-match (circles) and post-match (triangles). Dashed vertical lines mark |SMD| thresholds of 0.05 and 0.10; values <0.10 indicate adequate balance. Matching was 1:1 using Mahalanobis distance with 0.20 caliper, without replacement; SMDs used the pooled standard deviation.

## 3. Results

### 3.1. Cohort Characteristics

Between 2018 and 2023, a total of 236 patients in our department tested positive for *C. difficile* during perioperative screening using GDH immunoassays (CerTest Biotec^®^). Of these, 181 patients were included in this study. The cohort was divided into two subgroups: *C. difficile* colonization (CDC, n = 90) and *C. difficile* infection (CDI, n = 91). The mean age was 66.3 years, with an equal male-to-female ratio of 1:1.

### 3.2. CDC Group

Among the 90 CDC patients, 62 received antibiotic treatment and 28 did not. Surgical procedures were categorized into three main types: gastrointestinal, gynecological, and hepato-pancreato-biliary. The majority underwent gastrointestinal surgery (71.1%), followed by gynecological surgery (18.8%) and hepato-pancreato-biliary surgery (10.0%). A detailed distribution is shown in Table 1 and illustrated in Figure 3.

### 3.3. Preoperative Characteristics

Table 2 and Table 3 compare baseline covariates and preoperative characteristics—including gender, age, comorbidities, and laboratory values—between the two CDC subgroups before and after PSM. Ischemic heart disease (IHD) was the most prevalent comorbidity in both groups, observed in 74.2% of patients in the antibiotic group and 53.6% in the non-antibiotic group. Diabetes mellitus was the second most common, reported in 16.6% and 10.7% of patients, respectively. Preoperative laboratory parameters were similar between both groups: mean serum albumin levels were 4.26 g/dL in the antibiotic group and 4.27 g/dL in the non-antibiotic group; mean hemoglobin levels were 11.3 g/dL and 11.8 g/dL, respectively; mean C-reactive protein (CRP) levels were 37.5 mg/L and 11.8 mg/L; and mean serum creatinine levels were 0.97 mg/dL and 0.89 mg/dL, respectively.

### 3.4. Postoperative Outcomes

Postoperative outcomes before and after propensity score matching (PSM) are presented in Table 4 (pre-matching) and Table 5 (post-matching). Prior to matching, the antibiotic-treated group had a non-significant tendency towards a higher morbidity rate (32.2% vs. 21.4%), more surgical site infections (SSI) (9.6% vs. 0%), and a significantly longer length of hospital stay (14.1 vs. 10.4 days; *p* = 0.040). None of the CDC patients progressed to active *Clostridium difficile* infection. PSM was performed using age and comorbidities as matching variables to reduce selection bias and enhance comparability between groups. After PSM, the two groups were similar in terms of postoperative outcomes, including length of stay (*p* = 0.306). Postoperative laboratory parameters remained comparable across both groups. An odds ratio was calculated to assess the association between postoperative complications and whether patients received antibiotic treatment. The odds ratio for postoperative complications in antibiotic-treated CDC patients vs. non-treated was 1.7460 (95% CI 0.6122 to 4.9799, *p* = 0.2973). Table 6 and Figure 4 depict the influence of each variable in predicting postoperative morbidity. As shown, CDC antibiotic treatment had an AUC of 0.549 (95% CI 0.405–0.687), close to a random variable and lower than age, congestive heart failure, or diabetes. In the multivariable logistic regression of the unmatched cohort, prophylactic antibiotic use was not independently associated with postoperative complications (*p* = 0.855) (Figure 5) compared to other comorbidities.

## 4. Discussion

*Clostridium difficile* infection (CDI) remains a significant concern for surgeons, particularly in oncologic patients undergoing major abdominopelvic surgery. Given the immunocompromised state and frequent exposure to broad-spectrum antibiotics in this population, it is not uncommon for clinicians to initiate prophylactic antibiotic treatment even in CDC patients. However, the clinical value of such prophylactic strategies remains uncertain.

Our study sought to evaluate whether antibiotic treatment in CDC patients undergoing abdominopelvic oncologic surgery confers any protective effect in terms of postoperative outcomes. We showed that CDC patients who received prophylactic antibiotic treatments had similar outcomes compared to those who did not receive antibiotics. A non-significant trend (*p* > 0.05) toward higher postoperative morbidity (32.2% vs. 21.4%, *p* = 0.327) and surgical site infection (SSI) rates (9.6% vs. 0%, *p* = 0.101) was observed among CDC patients who received antibiotics. Following adjustment for confounding variables through propensity score matching (PSM), these differences were attenuated (morbidity: 30.7% vs. 23.0%; SSI: 11.5% vs. 0%) and remained statistically non-significant (morbidity: *p* = 0.538; SSI: *p* = 0.077). All patients who developed SSIs had undergone colorectal procedures where clean-contaminated surgical fields inherently carry higher infection risks. Additionally, while our study was not designed to assess microbiome alterations, the theoretical risk of gut dysbiosis and subsequent emergence of vancomycin-resistant Enterococcus (VRE) or other multidrug-resistant organisms (MDROs) should not be overlooked [18]. The literature has demonstrated that oral vancomycin is associated with colonization and expansion of vancomycin-resistant enterococci (VRE) in CDI-treated patients [19,20,21]. There seemed to be a longer LOS in the antibiotic-treated group (mean: 14.1 vs. 10.4 days; *p* = 0.040) before PSM, likely driven by the need to complete antibiotic regimens and isolation protocols.

Notably, none of our patients, regardless of antibiotic exposure, experienced *Clostridium difficile* infection (CDI) exacerbations during hospitalization. Our cohort comprised exclusively oncologic surgical patients undergoing abdominopelvic procedures, in contrast to a larger and more heterogeneous study—including individuals with hematologic malignancies, trauma, various oncologic conditions, and renal transplants—that reported an exacerbation rate of 25% [22]. Although the pathophysiology underlying CDI exacerbation remains incompletely understood, established contributing factors include microbiota disruption and immunosuppression [13,23]. Currently, no standardized protocol exists for the management of CDC patients. Several studies have investigated the potential benefit of probiotics in reducing CDI incidence; however, results remain inconsistent and inconclusive [24,25,26]. Given that hospital-acquired CDI remains the predominant source of transmission, rigorous preventive strategies are paramount. These include appropriate isolation of CDC patients, strict adherence to hand hygiene protocols by healthcare personnel, and meticulous antiseptic techniques [27].

Despite existing evidence and guidelines endorsing our findings that antibiotic prophylaxis in CDC is not required, the protocols refer to the general population including mostly low-risk immunocompetent patients. This has led practitioners to question whether in subgroups, such as immunosuppressed surgical oncological patients, low-dose Vancomycin could be used to reduce the risk of postoperative morbidity and CD exacerbation considering their intrinsic higher risk. Our results demonstrated no significant differences in postoperative outcomes between patients who received prophylactic antibiotics and those who did not, highlighting the importance of deferring prophylactic antibiotic use even in high-risk cancer patients colonized with CD.

It is important to recognize that asymptomatic *C. difficile* carriers remain at risk of developing symptomatic CDI if circumstances change; for example, if they are exposed to antibiotics or undergo major interventions. A large meta-analysis including 8725 patients demonstrated that carriers had a significantly higher risk of progressing to CDI than non-carriers. Specifically, patients colonized at hospital admission were 5.9 times more likely to develop CDI compared with non-colonized individuals (RR 5.86; 95% CI 4.21–8.16). The absolute risk of CDI among colonized patients was 21.8% (95% CI 7.9–40.1%) versus 3.4% (95% CI 1.5–6.0%) in non-colonized patients, with an attributable risk of 18.4% [28]. In practical terms, this translates into an approximate 10–20% conversion rate from colonization to infection, compared to less than 1–3% in non-carriers [29,30]. For patients preparing for major surgery or chemotherapy, this risk is particularly concerning, as these interventions may act as the catalyst that transforms benign colonization into clinically significant disease. Thus, while colonization itself is asymptomatic, its clinical relevance lies in its potential to precipitate CDI under stress and in its implications for infection control within hospitals.

When CDI does occur in the perioperative period, the consequences for oncologic patients can be substantial. Episodes of CDI frequently prolong hospitalization by days to weeks [7]. Instead of focusing on mobilization and recovery, patients may require bowel rest, intravenous fluids, and management of diarrhea or colitis. This shift in clinical course can delay wound healing and increase the risk of secondary complications such as central line infections or venous thromboembolism due to prolonged immobility.

In colorectal surgery, the impact of CDI is particularly concerning because of its potential to compromise anastomotic healing. The inflammatory response triggered by CDI can undermine anastomotic integrity. In one of our previous studies, the anastomotic leak rate among patients with postoperative CDI reached 22%, a 7.44-fold increase compared to controls (OR 7.44, 95% CI) [14]. Anastomotic leaks are among the most devastating surgical complications, often necessitating reoperation and stoma creation, and they are associated with delays in initiating adjuvant therapy—all of which can negatively affect long-term oncologic outcomes.

The timing of postoperative adjuvant therapy is critical for cancer prognosis. For many malignancies, including colorectal and ovarian cancer, initiating chemotherapy within 6–8 weeks after surgery is associated with optimal survival outcomes. However, patients who develop severe CDI may spend weeks in hospital recovering, potentially missing this therapeutic window. In a recent Korean study, ovarian cancer patients who developed CDI during chemotherapy experienced significant treatment interruptions, and some were unable to return to their baseline performance status, jeopardizing disease control [14].

A recent and noteworthy contribution supporting prophylactic treatment of asymptomatic *C. difficile* carriers is the STOP-CDI trial by Ziegler et al. [15]. This study evaluated a strategy of active screening for *C. difficile* colonization followed by targeted prophylaxis with enteral vancomycin in high-risk, immunocompromised inpatients, including solid-organ and stem cell transplant recipients, CAR-T patients, and individuals with hematologic malignancies. Between November 2021 and December 2023, 696 patients were screened, of whom 11.1% (77 patients) were colonized and received prophylaxis. Compared with historical, treatment-weighted controls, the intervention group showed a significant reduction in hospital-onset CDI (0.88% vs. 5.6%; OR 0.15, 95% CrI 0.06–0.30). Additional benefits included lower 90-day CDI incidence (OR 0.40, 95% CrI 0.25–0.64), reduced stool output (IRR 0.84, 95% CrI 0.77–0.92), and shorter hospital stays (–2.5 days, 95% CrI –3.4 to –1.5). Importantly, prophylaxis was not associated with an increase in vancomycin-resistant Enterococcus infections (OR 0.77, 95% CrI 0.33–1.75) or mortality (OR 0.44, 95% CrI 0.11–1.49). These results suggest that targeted prophylaxis in colonized, high-risk patients may reduce CDI and related morbidity without evident safety concerns.

Our hospital conducts routine screening for *Clostridium difficile* in all patients at the time of admission. This practice contrasts with the 2021 clinical guidelines from the Infectious Diseases Society of America (IDSA), the Society for Healthcare Epidemiology of America (SHEA), and the American College of Gastroenterology (ACG) [31,32] which clearly state that testing is not indicated in asymptomatic patients. Nevertheless, we consider our patient population to be at particularly high risk—comprising individuals with immunosuppression, oncological conditions, and/or multiple comorbidities. Furthermore, because we recognize that these patients are also highly vulnerable to postoperative complications, prophylactic antibiotics are frequently administered for extended periods. However, such prolonged and potentially inappropriate antibiotic use may contribute to both increased bacterial resistance and the risk of *Clostridium difficile* infection [33].

As noted, our study included a relatively small cohort of 90 patients. A more comprehensive investigation could be conducted if multiple hospitals tested patients for *Clostridium difficile* upon admission, identified carriers, and closely followed them to monitor any progression to active, symptomatic infection. Expanding the cohort would allow analysis not only across broad surgical categories but also by specific surgical procedures (e.g., anterior rectal resection, hysterectomy, total gastrectomy, etc.), with and without antibiotic treatment. Additionally, incorporating stool testing to assess microbiome changes before and after antibiotic exposure, with a non-antibiotic treatment group as a control, would provide valuable insights.

This study has its limitations. The retrospective nature of this study implies a higher risk of selection bias, although we did try to reduce its impact on the results by performing PSM in a one-to-one fashion, extracting comparable patients from the control group. The heterogeneity of the study group was low even before PSM, proven by the overall similar results between the initial groups and matched ones. Still, even with PSM, one cannot account for missing variables (i.e., nutritional status, intraoperative variables) which may represent additional confounding. Another limitation is the relatively small number of patients in the study groups which restricts the ability to detect smaller effect sizes and may explain non-significant trends. However, considering the relative low incidence of CDC in clinical practice, it is difficult to gather a larger cohort without multicenter collaboration. Another point of view is the variation in protocols among institutions. Perioperative antibiotic prophylaxis and infection control protocol vary and this reduces the external validity of our results. We should also consider that CD carriers are not homogenous groups as far as we can easily imagine different levels of colonization in the same category. There is also a limitation in the detection method and detection limit of the method in a rapid immunoassay test, as the rapid test (CerTest Biotec^®^) is advertised to have a sensitivity of 96.6% and specificity of 99.4%, but this may be lower, as stated by the producer, in the context of asymptomatic carriers or low colonization levels. The data provided by the producer are for symptomatic CDI.

## 5. Conclusions

Prophylactic antibiotic use in asymptomatic *Clostridium difficile*-colonized (CDC) patients undergoing abdominopelvic oncologic surgery demonstrated no clinical benefit in reducing postoperative complications or CD exacerbation. Despite limitations regarding the retrospective design of this study and potential residual confounding due to missing data, based on our results, the practice of offering prophylactic low-dose antibiotics in high-risk subgroups of CDC patients should be avoided. Further powered studies should clarify the above findings.

## Figures and Tables

**Figure 1 medicina-61-01606-f001:**
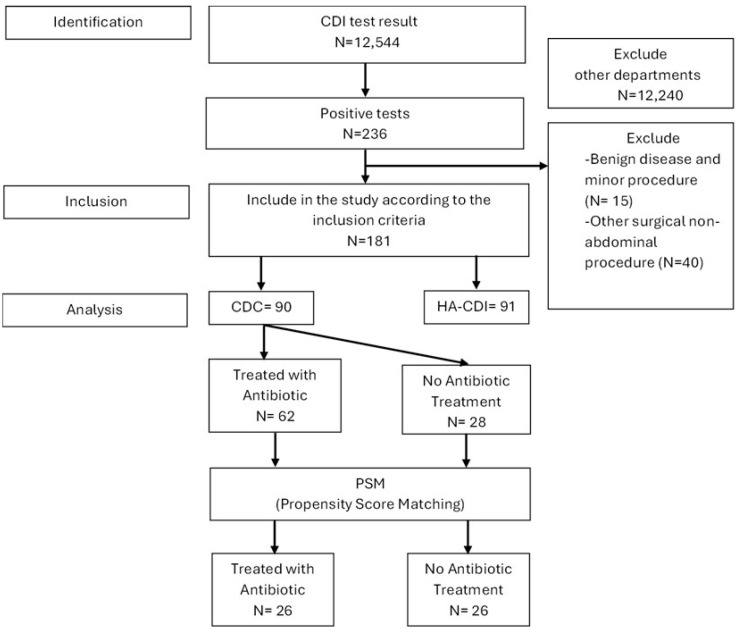
STROBE flowchart.

**Figure 2 medicina-61-01606-f002:**
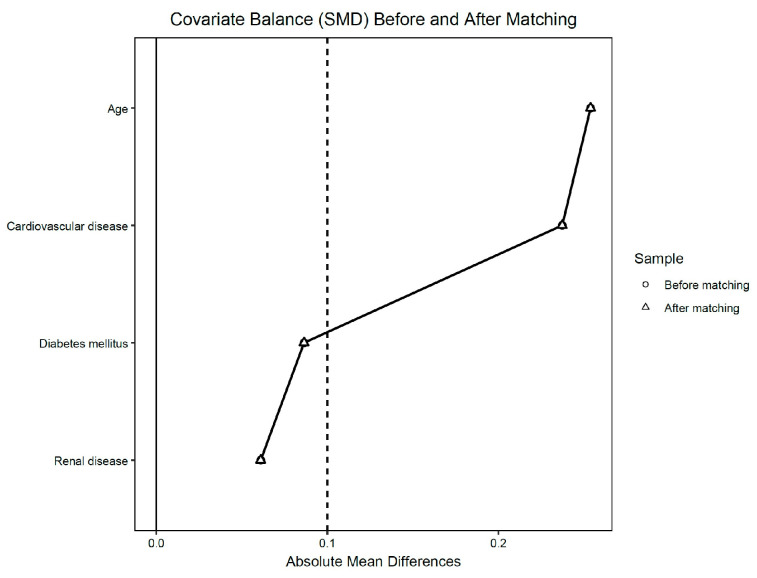
Love plot of covariate balance before and after matching.

**Figure 3 medicina-61-01606-f003:**
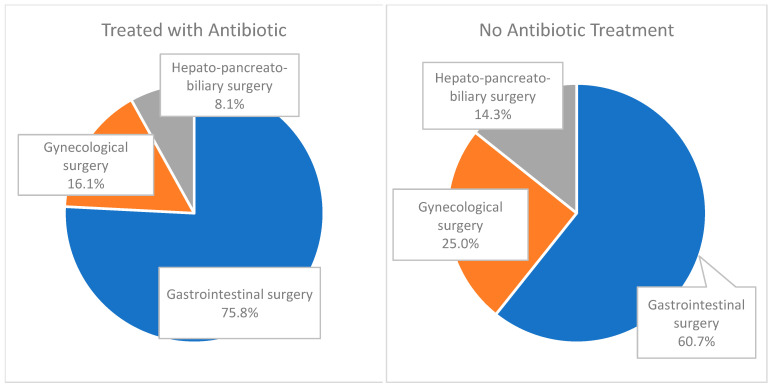
Distribution of surgical procedures between patients who received antibiotic treatment or not in CDC groups.

**Figure 4 medicina-61-01606-f004:**
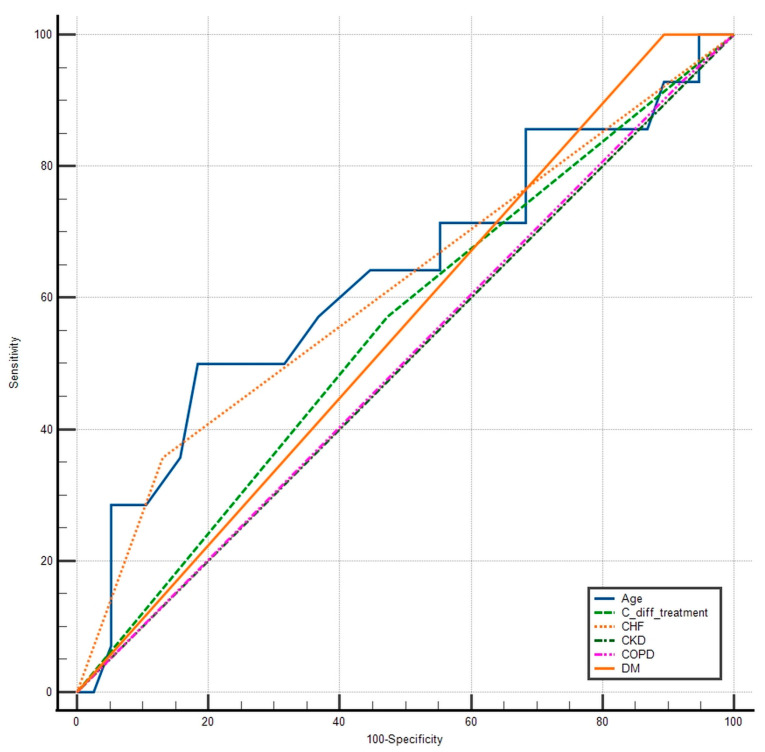
ROC curve comparison between the sensitivity and specificity of CDC antibiotic treatment, age, and comorbidities in predicting postoperative morbidity.

**Figure 5 medicina-61-01606-f005:**
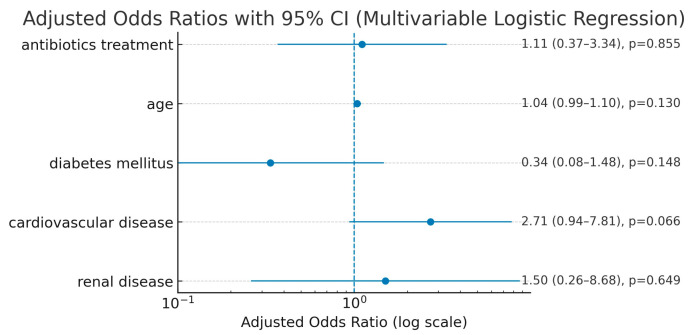
Forest plot of adjusted odds ratios (95% CI) from the multivariable logistic regression. Dots represent adjusted odds ratios; horizontal lines indicate 95% confidence intervals; the dashed reference line denotes OR = 1. *p*-values are shown at right; effects with CIs that do not cross 1 are considered statistically significant.

**Table 1 medicina-61-01606-t001:** Distribution of surgical procedures between patients who received antibiotic treatment or not in CDC groups.

	Treated with Antibiotic n (%)	No Antibiotic Treatment n (%)	Total
Total	62 (100)	28 (100)	90 (100)
Gastrointestinal surgery	47 (75.8)	17 (60.7)	64 (71.1)
Gynecological surgery	10 (16.1)	7 (25.0)	17 (18.8)
Hepato-pancreato-biliary surgery	5 (8.1)	4 (14.3)	9 (10.0)

**Table 2 medicina-61-01606-t002:** Comparison in terms of covariates and preoperative characteristics in CDC groups before PSM.

BEFORE PSM			
	Treated with Antibiotic n (%)/Mean (SD)	No Antibiotic Treatment n (%)/Mean (SD)	*p* Value
Total	62 (100)	28 (100)	
Gender (males)	32 (51.6)	9 (32.1)	*p* = 0.110
Age, mean (SD)	64.9 (12.8)	67.9 (10.0)	*p* = 0.224
Diabetes	12 (19.4)	3 (10.7)	*p* = 0.375
IHD	46 (74.2)	15 (53.6)	*p* = 0.086
COPD	5 (8.1)	0 (0)	*p* = 0.319
CKD	6 (9.7)	1 (3.6)	*p* = 0.428
PAD	0 (0)	0 (0)	-
Albumin (g/dL)	4.2 (0.7)	4.2 (0.5)	*p* = 0.512
Hb (g/dL)	11.3 (2.2)	11.8 (2.1)	*p* = 0.318
CRP (mg/dL)	37.5 (62.0)	49.8 (83.3)	*p* = 0.661
Creatinine (mg/dL)	0.9 (0.5)	0.8 (0.1)	*p* = 0.376

Key: IHD—Ischemic Heart Disease; COPD—Chronic Obstructive Pulmonary Disease; CKD—Chronic Kidney Disease; PAD—Peripheral Arterial Disease; Hb—Hemoglobin; CRP—C-Reactive Protein.

**Table 3 medicina-61-01606-t003:** Comparison in terms of covariates and preoperative characteristics in CDC groups after PSM.

AFTER PSM			
	Treated with Antibiotic n (%)/Mean (SD)	No Antibiotic Treatment n (%)/Mean (SD)	*p* Value
Total	26 (100)	26 (100)	
Gender (males)	16 (61.5)	9 (34.6)	*p* = 0.035
Age, mean (SD)	64 (11.5)	64.1 (12.6)	*p* = 0.973
Diabetes	2 (7.6)	2 (7.6)	*p* = 1.000
IHD	14 (53.8)	13 (50.0)	*p* = 0.781
COPD	4 (15.3)	0 (0)	*p* = 0.037
CKD	0 (0)	0 (0)	-
PAD	0 (0)	0 (0)	-

Key: IHD—Ischemic Heart Disease; COPD—Chronic Obstructive Pulmonary Disease; CKD—chronic kidney disease; PAD—Peripheral Arterial Disease.

**Table 4 medicina-61-01606-t004:** Comparison in terms of covariates and postoperative outcomes in CDC group before PSM.

Before PSM			
	Treated with Antibiotic n (%)/Mean (SD)	No Antibiotic Treatment n (%)/Mean (SD)	*p* Value
Total	62 (100)	28 (100)	
Morbidity	20 (32.2)	6 (21.4)	*p* = 0.327
Exacerbation	0 (0)	0(0)	-
Mortality	2 (3.22)	0 (0)	*p* = 1.000
SSI	6 (9.6)	0 (0)	*p* = 0.171
Anastomotic Leak	1(1.6)	0 (0)	-
LOS	14.1 (8.9)	10.4 (4.3)	*p* = 0.040
Albumin (g/dL)	2.9 (0.61)	3.2 (0.7)	*p* = 0.546
Hb (g/dL)	10.2 (1.7)	10.6 (1.6)	*p* = 0.425
CRP (mg/dL)	95.7 (61.5)	61.5 (74.5)	*p* = 0.195
NLR	8.3 (9.7)	7.2 (5.7)	*p* = 0.619
Creatinine (mg/dL)	0.9 (0.6)	0.7 (0.2)	*p* = 0.127

Key: SSI—Surgical Site Infection; LOS—Length of Hospital Stay; Hb—Hemoglobin; CRP—C-Reactive Protein; NLR—Neutrophil-to-Lymphocyte Ratio.

**Table 5 medicina-61-01606-t005:** Comparison in terms of covariates and postoperative outcomes in CDC group after PSM.

After PSM			
	Treated with Antibiotic n (%)/Mean (SD)	No Antibiotic Treatment n (%)/Mean (SD)	*p* Value
Total	26 (100)	26 (100)	
Morbidity	8 (30.7)	6 (23.0)	0.538
Mortality	1 (3.8)	0 (0)	0.317
SSI	3 (11.5)	0 (0)	0.077
Anastomotic leak	0 (0)	0 (0)	-
LOS	11.7(3.9)	10.5(4.5)	0.306
Hb (g/dL)	10.6(1.8)	10.6(1.6)	0.917
CRP (mg/dL)	83.0 (59.6)	61.5 (74.5)	0.431
NLR	6.4 (4.0)	7.1 (5.9)	0.597
Creatinine (mg/dL)	0.8 (0.2)	0.7 (0.2)	0.443

Key: SSI—Surgical Site Infection; LOS—Length of Hospital Stay; Hb—Hemoglobin; CRP—C-Reactive Protein; NLR—Neutrophil-to-Lymphocyte Ratio.

**Table 6 medicina-61-01606-t006:** ROC curve analysis comparing AUC values in predicting postoperative morbidity between patients that received CDC treatment, age, and comorbidities.

Variable	AUC	SE	95% CI
Age	0.631	0.0957	0.486 to 0.760
*C. diff* treatment	0.549	0.0800	0.405 to 0.687
IHD	0.613	0.0720	0.468 to 0.745
CKD	0.500	0.000	0.358 to 0.642
COPD	0.504	0.0420	0.362 to 0.645
DM	0.553	0.0252	0.408 to 0.691

Key: *C. diff*—*Clostridium difficile*; IHD—Ischemic Heart Disease; CKD—Chronic Kidney Disease; COPD—Chronic Obstructive Pulmonary Disease; DM—Diabetes Mellitus.

## Data Availability

The data that support the findings of this study are available on request from the corresponding author, WLO.

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
