# Peer review of "Clostridium difficile* Colonization in Oncologic Patients Undergoing Major Abdominopelvic Surgery: To Treat or Not to Treat? An Observational Study with Propensity Score Analysis"

_medicina, 2025, doi:10.3390/medicina61091606_

Round 1

Reviewer 1 Report

Comments and Suggestions for Authors

This is an interesting study and the authors have collected a unique dataset using cutting edge methodology.

The article is a retrospective single-center observational study stated that prophylactic antibiotics use in Clostridium difficile colonization (CDC) patients undergoing abdominopelvic oncology surgery offer no clear clinical benefit when comparing to non-colonized control.

The authors employed propensity score matching (PSM) to compare each exposure group to matched controls. The findings have important clinical implications suggesting that asymptomatic colonization may not contraindicate elective surgery, whereas postoperative infection demands urgent attention.

The topic is clinically relevant, addressing an important perioperative infection control question with implications for surgical oncology practice.

The manuscript is generally well written, but the statistical reporting and methodological transparency require strengthening.

Overall, the information presented represents valuable information regarding the feasibility of using prophylactic antibiotics in CDC patients undergoing abdominopelvic oncology surgery that offers no clear clinical benefit and may contribute to antimicrobial resistance  

However, in my opinion the paper has some shortcomings in regards to some data analyses and text, and I feel this unique dataset has not been utilized to its full extent.

My Major and Minor Comments

  • The Propensity Score Matching used in the manuscript does not specify covariates included in the PSM, the matching algorithm used, caliper width, or whether matching was performed with or without replacement.
  • The Balance Diagnostics is Absent There is no reporting of standardized mean differences (SMDs) or graphical diagnostics before and after matching.

  • While PSM reduces bias, retrospective data cannot fully adjust for unmeasured confounders (e.g., nutritional status, intraoperative variables).
  • The Small CDC and CDI group sizes limits statistical power and precision, particularly in subgroup comparisons and detecting smaller effect sizes.

  • External validity may be limited by center-specific protocols (antibiotic prophylaxis, perioperative care).

  • If some CDC patients received preoperative treatment, this could influence outcomes; details of management strategies would be helpful.
  • Focus is on short-term postoperative outcomes; data on readmission, recurrence, or long-term complications would add value.
  • Discussion could be enriched by referencing other studies discussing the role of prophylactic antibiotics in patients undergoing major abdominopelvic operations.
  • The authors did not give any data about the type of surgical procedures as it is documented that in some colonic surgeries anastomotic leak can occurs which can increase the morbidity due to CDI.
  • The numbers and percentage of cases presented in tables and figures should be revised. (Table 1 and 2 and Figure 2)
  • All abbreviations are to be written in full when they first appear in text.
  • Use consistent terms (e.g., “CDC” vs “colonized” vs “asymptomatic carrier”).
  • Propose future directions to suggest prospective multicenter studies or trials examining prophylactic strategies for CDC patients or interventions minimizing CDI risk.

Comments on the Quality of English Language

The English could be improved to more clearly express the research.

Author Response

Dear Reviewer,

We appreciate the time and effort that you dedicated to our study, and we are grateful for the interesting comments and valuable suggestions you have made to the paper.

In this revised manuscript we have incorporated most of the comments and highlighted the changes to the text in red. Please find below our point-by-point replies and changes to your suggestions.

Comment 1: The Propensity Score Matching used in the manuscript does not specify covariates included in the PSM, the matching algorithm used, caliper width, or whether matching was performed with or without replacement.

Answer: We appreciate this observation. In the revised manuscript, we have now specified that PSM was performed using age and comorbidity burden as covariates. Matching was conducted in a 1:1 ratio without replacement, using Mahalanobis distance matching with a caliper of 0.2. This information has been added to the Materials and Methods section (page 4, lines 149–158).

Comment 2: The Balance Diagnostics is Absent There is no reporting of standardized mean differences (SMDs) or graphical diagnostics before and after matching.

Answer: Thank you for this important suggestion. We have now depicted standardized mean differences (SMDs) before and after matching in a Love plot (Figure 2) (page 5, lines 171-178)

Comment 3: While PSM reduces bias, retrospective data cannot fully adjust for unmeasured confounders (e.g., nutritional status, intraoperative variables).

Answer: We agree and have acknowledged this limitation explicitly in the Discussion section (page 13, lines 385–390), highlighting that the above and other variables were not available and may represent residual confounding.

Comment 4: The Small CDC and CDI group sizes limits statistical power and precision, particularly in subgroup comparisons and detecting smaller effect sizes.

Answer: Thank you for your suggestion. We stated this limitation in the Discussion noting that the relatively small sample size of both CDC and CDI groups restricts the ability to detect smaller effect sizes and may explain non-significant trends (page 13, lines 389-392).

Comment 5: External validity may be limited by center-specific protocols (antibiotic prophylaxis, perioperative care).

Answer: We agree. We have clarified in the Discussion that our findings should be interpreted with caution, as perioperative antibiotic prophylaxis and infection control protocols may vary across institutions. This limitation has been highlighted (page 13, lines 393–395).

Comment 6: If some CDC patients received preoperative treatment, this could influence outcomes; details of management strategies would be helpful.

Answer: Thank you for pointing this out. We clarified in the Methods that antibiotic treatment in CDC patients was exclusively oral vancomycin (125 mg QDS), initiated perioperatively at the discretion of the attending consultant. No patients received systemic preoperative antibiotics beyond standard prophylaxis. This is our local protocol and to ensure adherence to the protocol, for this study two authors checked the drug Kardex of each of the included patients to verify if another antibiotic was used. This has been specified in the revised text (page 3, lines 116–120).

Comment 7: Focus is on short-term postoperative outcomes; data on readmission, recurrence, or long-term complications would add value.

Answer: We acknowledge this limitation. Unfortunately, data on readmission, recurrence, or long-term complications was beyond the purpose of this study. We might redo the database including these endpoints in a future study.

Comment 8: Discussion could be enriched by referencing other studies discussing the role of prophylactic antibiotics in patients undergoing major abdominopelvic operations.

Answer: Thank you for this suggestion. In the revised version we have added a couple of paragraphs in the Discussion section highlighting the above (pages 12-13, lines 313-375)

Comment 9: The authors did not give any data about the type of surgical procedures as it is documented that in some colonic surgeries anastomotic leak can occur which can increase the morbidity due to CDI.

Answer: Thank you for your suggestion. We have split the two groups in terms of procedures in Table 1 and Figure 3. We have also added a new paragraph in the Discussion section commenting on the risk of anastomotic leakage in CDC patients (page 12, lines 334-341). In fact, we have published a previous study showing that CDC patients undergoing colorectal surgery do not have a higher risk of anastomotic leak, however HA-CDI (hospital acquired CDI) have a seven times higher risk of anastomotic leak. In this revised version we have added data on anastomotic leaks in Tables 4 and 5 (before and after PSM).

Comment 10: The numbers and percentage of cases presented in tables and figures should be revised. (Table 1 and 2 and Figure 2)

Answer: Thank you for your request for revision. Upon rechecking the data, we identified that the only correction needed in Table 2 concerns the percentage of patients with diabetes who received antibiotic treatment, which should be revised from 16.6% to 19.4%. For Table 1 and Figure 2, we have ensured consistency in the reported percentages for each type of surgical procedure.

Comment 11: All abbreviations are to be written in full when they first appear in text.

Answer: We have amended this.

Comment 12: Use consistent terms (e.g., “CDC” vs “colonized” vs “asymptomatic carrier”).

Answer: We have amended this.

Comment 13: Propose future directions to suggest prospective multicenter studies or trials examining prophylactic strategies for CDC patients or interventions minimizing CDI risk.

Answer: We appreciate your valuable suggestions. An additional paragraph has been included in the Discussion section to propose future directions (page 13, lines 376-384).

Reviewer 2 Report

Comments and Suggestions for Authors

Although the manuscript has important clinical implications with encouraging results, several issues need to be addressed:

Describe the C. difficile testing protocol in detail including well-defined case definitions, e.g., type of test used to detect infection (e.g. enzyme immunoassay, PCR), sensitivity/specificity for colonization versus infection and laboratory confirmation. cite doi:10.1093/cid/ciad606. 

Clinicians can be further educated on what criteria should warrant the use of prophylactic antibiotics.

Low numbers, especially amongst non-antibiotic group (n=28), limits statistical power

The covariates which were selected for propensity score matching should be justified by the authors.

The use of ROC curve analysis is correct, however it may be improved with multivariate regression to account for potential confounders

Results presentation issues:

Figure 2. This graph is somewhat misleading as it gives the effect that there is a difference between groups, when in fact there is not Table 3 Showing no subgroup by treatment interaction.

Table 1 Would Benefit from Per Cent of Total Procedures in Each Category

The authors should include a table that displays the baseline characteristics after propensity score matching indicating the success of the matching.

The authors recognize the limitation of a small sample size and should address how this affects the interpretation of their findings

Selection bias in identifying which patients received antibiotics prophylaxis deserves special consideration

The question should focus on the reasoning behind selection of oral vancomycin as the prophylactic. discuss and cite the global burden of multiresistant bacteria and implications on prognosis and healthcare system. cite doi:10.3390/epidemiologia6020021 

The appropriate conclusion that differences in mortality suggest prophylactic antibiotics should be avoided, should be set in the context of the study limitations

The authors should comment on how their findings are consistent with existing C. difficile colonization management guidelines

Several typos, missing words and other grammatical issues that make the article a bit harder to read

The abstract needs to be restructured to emphasize key findings more thoroughly

Including the newest literature on C. difficile colonization management in references

This study could be an important addition to clinical practice, if concerns have been resolved by a proper revision, challenging the standard use of prophylactic antibiotics in C. difficile colonized oncologic patients undergoing major abdominopelvic surgery.

Author Response

Dear Reviewer,

We appreciate the time and effort that you dedicated to our study, and we are grateful for the interesting comments and valuable suggestions you have made to the paper.

In this revised manuscript we have incorporated most of the comments and highlighted the changes to the text in red. Please find below our point-by-point replies and changes to your suggestions.

Comment 1: Describe the C. difficile testing protocol in detail including well-defined case definitions, e.g., type of test used to detect infection (e.g. enzyme immunoassay, PCR), sensitivity/specificity for colonization versus infection and laboratory confirmation. cite doi:10.1093/cid/ciad606. 

Answer: Thank you for your suggestion. A more detailed definition of colonization versus infection, as well as the sensitivity and specificity of the rapid test, has been added to the Materials and Methods section (page 3, lines 97–109).

Comment 2: Clinicians can be further educated on what criteria should warrant the use of prophylactic antibiotics.

Answer: Thank you for your suggestion. An additional paragraph has been included in the Discussion to aid in decision-making regarding prophylactic antibiotic administration (page 13, lines 363-373).

Comment 3: Low numbers, especially amongst non-antibiotic group (n=28), limits statistical power

Answer: Thank you for your suggestion. We stated this limitation in the Discussion noting that the relatively small sample size of both CDC and CDI groups restricts the ability to detect smaller effect sizes and may explain non-significant trends (page 13, lines 383-400).

Comment 4: The covariates which were selected for propensity score matching should be justified by the authors.

Answer: We appreciate this observation. In the revised manuscript, we have now specified that PSM was performed using age and comorbidity burden (diabetes, respiratory diseases, cardiovascular diseases and renal disease) as covariates. Matching was conducted in a 1:1 ratio without replacement, using Mahalanobis distance matching with a caliper of 0.2. This information has been added to the Materials and Methods section (page 4, lines 151-160).

Comment 5: The use of ROC curve analysis is correct, however it may be improved with multivariate regression to account for potential confounders

Answer: We agree and have added a multivariable logistic regression in the unmatched cohort adjusting for age, comorbidities, sex, and surgery type as a sensitivity analysis. The adjusted association between prophylactic antibiotics and postoperative complications remained non-significant and directionally consistent with the primary analysis; full model coefficients are provided in Figure 5.

Comment 6: Figure 2. This graph is somewhat misleading as it gives the effect that there is a difference between groups, when in fact there is not Table 3 Showing no subgroup by treatment interaction.

Answer: Thank you for this remark. We have amended Figure 2 with percentages to capture a better picture of the difference in procedures done in each group.

Comment 7: Table 1 Would Benefit from Per Cent of Total Procedures in Each Category

Answer: The total percentages are presented in Table 1: 71.1% of CDC patients underwent gastrointestinal surgery, 18.8% gynecological surgery, and 10.0% hepatopancreatobiliary surgery.

Comment 8: The authors should include a table that displays the baseline characteristics after propensity score matching indicating the success of the matching.

Answer: Thank you. We have added a new table (Table 3, page 7, lines 214-218)

Comment 9: The authors recognize the limitation of a small sample size and should address how this affects the interpretation of their findings

Answer: Thank you for your suggestion. We stated this limitation in the Discussion noting that the relatively small sample size of both CDC and CDI groups restricts the ability to detect smaller effect sizes and may explain non-significant trends (page 13, lines 383-400).

Comment 10: Selection bias in identifying which patients received antibiotics prophylaxis deserves special consideration

Answer: We have clarified that antibiotic initiation for CDC was consultant-driven based on perceived risk of CDI exacerbation. We emphasize that PSM was used to mitigate confounding by indication, but residual selection bias may remain (page 13, lines 383-400)

Comment 11: The question should focus on the reasoning behind selection of oral vancomycin as prophylactic. discuss and cite the global burden of multiresistant bacteria and implications on prognosis and healthcare system. cite doi:10.3390/epidemiologia6020021 

Answer: Thank you. In the revised manuscript we have added two more paragraphs discussing the implication of antibiotic prophylaxis (pages 12-13, lines 311-373)

Comment 12: The appropriate conclusion that differences in mortality suggest prophylactic antibiotics should be avoided, should be set in the context of the study limitations

Answer: Thank you for your suggestion. The Conclusion now explicitly states that findings do not support prophylactic antibiotics for CDC in oncologic abdominopelvic surgery and should be interpreted in light of the retrospective design, small sample, and potential residual confounding.

Comment 13: The authors should comment on how their findings are consistent with existing C. difficile colonization management guidelines

Answer: Thank you, we have extensively revised our Discussion section and addressed this point and the above suggestions.

Comment 14: Several typos, missing words and other grammatical issues that make the article a bit harder to read

Answer: We thank the reviewer for highlighting these issues. The manuscript has been carefully revised to correct typos, fill in missing words, and address grammatical errors to improve clarity and readability throughout the text.

Comment 15: The abstract needs to be restructured to emphasize key findings more thoroughly

Answer: We appreciate this valuable suggestion. The abstract has been restructured to emphasize the key findings more clearly and to improve the overall clarity and impact of the summary.

Comment 16: Including the newest literature on C. difficile colonization management in references

Answer: Both the 2021 clinical guidelines from the Infectious Diseases Society of America (IDSA) and the Society for Healthcare Epidemiology of America (SHEA), as well as those from the American College of Gastroenterology (ACG) have been discussed and cited in the revised manuscript.

Round 2

Reviewer 1 Report

Comments and Suggestions for Authors

We appreciate your reply to my comments and  suggestions you have made to the paper.

I revised manuscript and I found that you have incorporated most of the comments that can improve the final version.